# Assessment of a Hybrid Eulerian–Lagrangian CFD Solver for Wind Turbine Applications and Comparison with the New MEXICO Experiment

Nikos Spyropoulos [1,*], George Papadakis [2], John M. Prospathopoulos [1] and Vasilis A. Riziotis [1,*]

1 School of Mechanical Engineering, National Technical University of Athens, GR15780 Athens, Greece
2 School of Naval & Marine Engineering, National Technical University of Athens, GR15780 Athens, Greece
* Correspondence: nspyro@fluid.mech.ntua.gr (N.S.); vasilis@fluid.mech.ntua.gr (V.A.R.);
Tel.: +30-210-772-1097 (N.S.); +30-210-772-1101 (V.A.R.)

**Abstract:** In this paper, the hybrid Lagrangian–Eulerian solver HoPFlow is presented and evaluated against wind tunnel measurements from the New MEXICO experiment. In the paper, the distinct solvers that assemble the HoPFlow solver are presented, alongside with details on their mutual coupling and interaction. The Eulerian solver, MaPFlow, solves the compressible Navier–Stokes equations under a cell-centered finite-volume discretization scheme, while the Lagrangian solver uses numerical particles that carry mass, pressure, dilatation and vorticity as flow markers in order to represent the flow-field by following their trajectories. The velocity field is calculated with the use of the decomposition theorem introduced by Helmholtz. Computational performance is enhanced by utilizing the particle mesh (PM) methodology in order to solve the Poisson equations for the scalar potential $\phi$ and the stream function $\vec{\psi}$. The hybrid solver is tested in 3-D unsteady simulations concerning the axial flow around the wind turbine (WT) model rotor tested in the New MEXICO experimental campaign. Simulation results are presented as integrated rotor loads, radial distribution of aerodynamic forces and moments and pressure distributions at various span-wise positions along the rotor blades. Comparison is made against experimental data and computational results produced by the pure Eulerian solver. A total of 5 PM nodes per chord length of the blade section at 75% have been found to be sufficient to predict the loading at the tip region of the blade with great accuracy. Discrepancies with respect to measurements, observed at the root and middle sections of the blade, are attributed to the omission of the spinner geometry in the simulations.

**Keywords:** turbine aerodynamics modelling; hybrid CFD solvers; vortex particles; particle mesh; New MEXICO

## 1. Introduction

Wind energy is a substantial paradigm of human technology in which atmospheric air flows are leveraged for environmentally neutral energy generation. In the effort to reduce wind energy costs, the wind energy sector is directed towards increasing the size and flexibility of modern wind turbine (WT) rotors. To that end, rotor blades are becoming larger and more slender, leading to high complexity regarding the analysis of their loading and the description of the flow field in their wake. In connection to the above need for highly flexible designs, rotor analysis is today directed towards high fidelity numerical tools, applying elaborated, physically motivated models for the accurate prediction of the response of the rotor blades. This requires the use of highly accurate computational models both for the elasto–dynamic and the aerodynamic analysis of the blades, which, combined together, comprise the field of aeroelasticity.

In terms of aerodynamic analysis, computational fluid dynamics (CFD) methodologies are considered to be the most common option among several numerical approaches in terms of accuracy. The advantage of employing CFD in rotor analysis is related to its capability

to account in maximum detail for viscous and turbulence effects. These are dominant in wind farm simulations, where rotor wake evolution has an impact on the performance of downwind turbines. Furthermore, viscous effects are important when studying the rotor wake interaction with the boundary layer developed on surrounding bodies, such as the WT tower or even the ground. Such interactions affect the rotor performance and the blades' loading.

Despite being accurate and reliable, the increased computational cost of conventional Eulerian CFD methodologies renders them prohibitive for aeroelastic design simulations. As a remedy, research is today directed towards the development of novel CFD methodologies that share the same level of accuracy as the conventional methodologies under reduced computational requirements. In this direction, domain decomposition seems to be an attractive technique provided that the best performing formulation in terms of cost and accuracy is employed in each sub-domain. The proposed hybrid CFD methodology combines a standard finite volume Eulerian CFD approach close to the solid boundaries with a Lagrangian CFD approach in vorticity formulation for the rest of the domain [1–3].

The flow–field may be discretised in two different ways, thus classifying CFD methodologies into Eulerian and Lagrangian methodologies. On the one hand, Eulerian approaches discretise the entire computational domain in "stationary" nodes through which the fluid moves and on whom the flow properties are recorded. On the other hand, in Lagrangian methods, the fluid is discretised in "numerical particles". In this way, the flow-field is described in a material approach by tracking the motion and following the flow properties of the fluid particles.

Eulerian methodologies [4–6] have gained much popularity and reliability, thanks to the accurate consideration of solid-wall boundaries, as no-penetration and no-slip conditions are both satisfied in maximum detail. However, this is not the case regarding the far-field boundaries. Truncation of the flow at a finite distance, where the boundary conditions typically approximate those at infinity, is a numerical approximation that leads to errors and is considered as significant drawback in external aerodynamics applications. Furthermore, domain truncation is usually combined with gradual grid coarsening, which increases numerical diffusion and adds errors as well. This may be catastrophic in wind farm applications, where a detailed description of the wake of the leading WTs is crucial for the correct estimation of the total power production. Local grid refinement seems like an obvious remedy, but it comes with a substantial increase in computational cost. Moreover, the consideration of the interaction between independently moving bodies is usually implemented through the application of special methodologies (e.g., overset, chimera or sliding grids), which penalize computational cost as well.

The alternative would be to solve the Navier–Stokes equations in the Lagrangian formulation, as particle methods do. They are grid–free, self–adaptive and have (in theory) zero numerical diffusion. The intuitive option would be to define particles that carry mass, momentum and energy, as the smooth particle hydrodynamic (SPH) methods do [7–9]. Alternatively, vorticity [10,11] and dilatation [12,13] may be used as the primary flow variables leading to vortex methodologies (VM). Due to their robustness in pressure variations, VM are quite popular in external aerodynamics applications [14–16]. Consequently, they are considered suitable for WT applications [17–21] as well. A major challenge for particle methods is the treatment of solid-wall boundaries [22,23]. Another drawback is the fact that the computational cost rises proportionally to $N^2$ ($N$ is the number of particles), thus rendering long–lasting simulations computationally prohibitive. To overcome this problem, methodologies such as the particle-mesh (PM) [24] may be adopted in order to enhance performance.

To sum up, Lagrangian methodologies are more effective in the far-field region, whereas Eulerian methodologies are considered to be ideal for the region close to the solid-wall boundaries. It is therefore reasonable to combine the two methodologies through a domain decomposition technique in order to enhance accuracy and reduce computational cost. In general, the sub-domains may either overlap or not. Strong viscous-inviscid interaction models [25,26] and RANS-vortex coupled models [27,28] are examples of completely overlapping hybrid methodologies, while the method presented in [29] is an example of limited overlapping over a buffer area.

In the current project, an in-house hybrid solver, named HoPFlow, is presented and assessed in WT applications. A preliminary version of the solver and results concerning low subsonic turbulent flows around airfoils are presented in [1]. Implementation details have been presented analytically in [3], where compressibility effects have been also taken into account. Recently, the hybrid solver has been used in $3D$, WT applications where an axial flow test case of the New MEXICO experimental campaign [30,31] was simulated. Preliminary results have been presented in [32], focusing on aerodynamic load prediction and near-wake flow field computations. In this study, the numerical parameters of the presented solver that affect load predictions (e.g., blade surface mesh, time-step value, PM discretisation length) are analyzed in detail. The main motivation of the present work is to validate the flow-field characteristics close to solid boundaries using measured datasets and thus assess the coupling procedure applied in the hybrid solver. As indicated in the discussion above, it is noted that the value of the present method lies in applications in which there are multiple, mutually interacting regions of interest within the computational space [33] (e.g., multiple-rotor configurations). The Lagrangian domain serves as the coupling domain between these regions of interest in the same way that overset grids function in standard Eulerian CFD simulations. Lagrangian methods are advantageous compared to overset approaches, as they minimize numerical diffusion and therefore preserve wake structures and flow disturbances.

The test case is the run no. 266, which is a 14.7 m/s axial flow case at 425 rpm of the New MEXICO experimental campaign [30,31] that corresponds to a tip speed ratio of $\lambda$ = 6.81 and a pitch angle of 2.3° nose down. The New MEXICO WT model rotor is a 4.5 m diameter 3–bladed rotor. It consists of three different airfoil shapes at the root (DU91-W2-250), mid-span (RISØ A1-21) and tip (NACA 64418) region of the blade according to Table 1. The twist and chord distribution of the blade is shown in Table 2. Turbulent transition is triggered with trip tapes of 5 mm width and 0.2 mm thickness placed at 10% chord of both pressure and suction sides of the blade. A thorough comparison of blade loads is performed against experimental measurements and computational results from the Eulerian counterpart (MaPFlow) of the hybrid solver. Blade loads are depicted as integrated rotor loads, span–wise distribution of aerodynamic loads and pressure distribution at specific span–wise positions. More results from numerical investigations concerning test cases from the MEXICO and the New MEXICO experimental campaigns can be found in [34–40].

**Table 1.** Mexico rotor blade airfoils.

| Radius [m] | r/R | Airfoil |
|---|---|---|
| 0.45–1.025 | 20–45% | DU91-W2-250 |
| 1.225–1.475 | 55–65% | RISØ A1-21 |
| 1.675–2.25 | 75–100% | NACA 64418 |

**Table 2.** Mexico rotor blade twist and chord distribution.

| Radius [m] | Twist [°] | Chord [mm] |
|:---:|:---:|:---:|
| 0.21 | 0 | 195 |
| 0.23 | 0 | 195 |
| 0.235 | 0 | 90 |
| 0.300 | 0 | 90 |
| 0.375 | 8.2 | 165 |
| 0.450 | 16.4 | 240 |
| 0.675 | 12.1 | 207 |
| 0.900 | 8.3 | 178 |
| 1.025 | 7.1 | 166 |
| 1.125 | 6.1 | 158 |
| 1.225 | 5.5 | 150 |
| 1.350 | 4.8 | 142 |
| 1.475 | 4.0 | 134 |
| 1.575 | 3.7 | 129 |
| 1.675 | 3.2 | 123 |
| 1.800 | 2.6 | 116 |
| 2.025 | 1.5 | 102 |
| 2.165 | 0.7 | 92 |
| 2.193 | 0.469 | 82 |
| 2.232 | 0.231 | 56 |
| 2.250 | 0.0 | 11 |

## 2. Methodology

The rationale behind the application of the hybrid CFD solver HoPFlow is to combine an Eulerian approach close to solid-wall boundaries with a Lagrangian one for the rest of the domain. In this way, both the solid-wall and the exact far-field boundary conditions are satisfied in maximum accuracy within the Eulerian and Lagrangian framework, respectively. The Lagrangian particles are distributed over the whole computational domain, overlapping with the Eulerian computational cells close to solid boundaries (see Figure 1). The Eulerian part of the hybrid solver solves the compressible Navier–Stokes equations under a cell-centered finite-volume approach in a confined region ($D_E$) around solid-wall boundaries ($S_B$). The Lagrangian part solves the compressible flow equations as well, in their material form, based on particle representation of the essential flow quantities, e.g., mass, pressure, dilatation and vorticity [12]. Lagrangian particles are distributed over the entire computational space and, hence, inside the Eulerian domain as well. The coupling procedure consists of two parts: (i) the Lagrangian particles solution is interpolated to the ghost cells of the Eulerian domain to define the correct boundary conditions on its far-field $S_E$ (see Figure 2); (ii) the Eulerian solution is used in order to correct the solution of the Lagrangian particles that lie within the Eulerian domain (see Figure 3). The corrected Lagrangian particles information is then used in order to update the whole Lagrangian field and thus ensure that it will be a smooth extension of the Eulerian one. More details concerning the coupling procedure are given in Section 2.3.

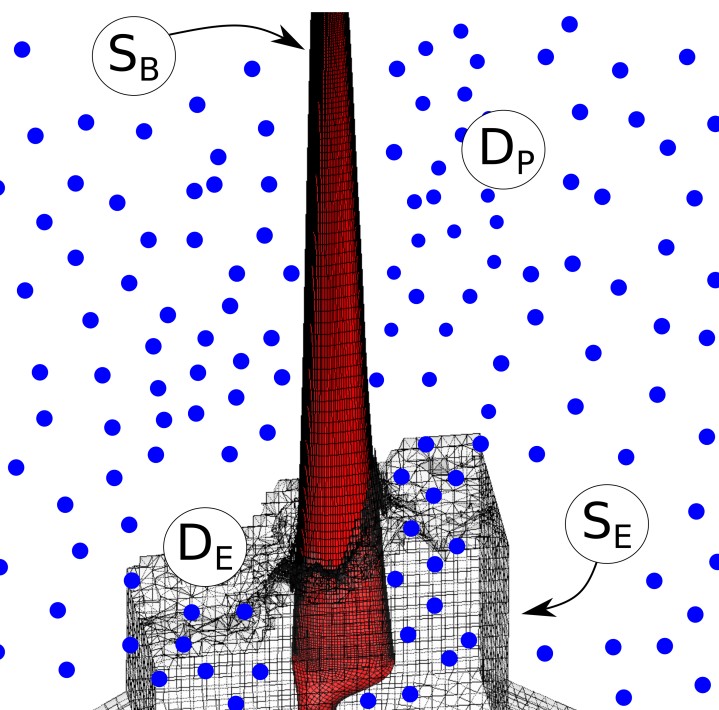

**Figure 1.** Decomposition of Eulerian ($D_E$) and Lagrangian ($D_P$) computational domains. $S_B$ denotes the solid-wall boundaries, and $S_E$ the far-field of the Eulerian domain.

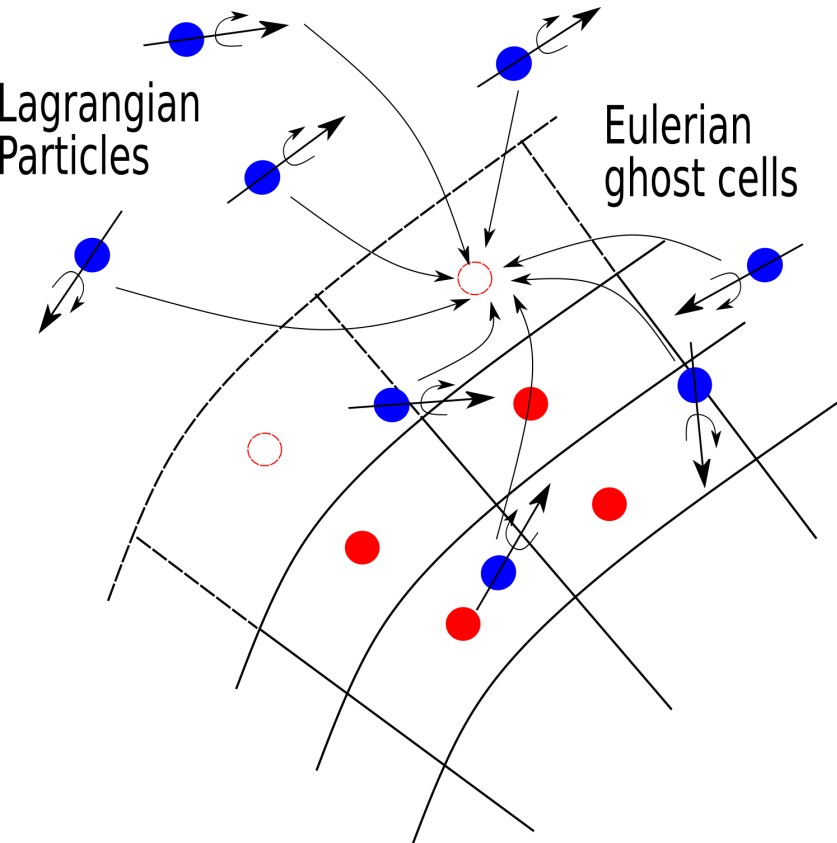

**Figure 2.** The Lagrangian particle solution is used to define the far-field boundary conditions of the Eulerian solver. Lagrangian particles are depicted as solid blue circles. Solid and dotted red circles denote the centers of the Eulerian cells and ghost cells, respectively.

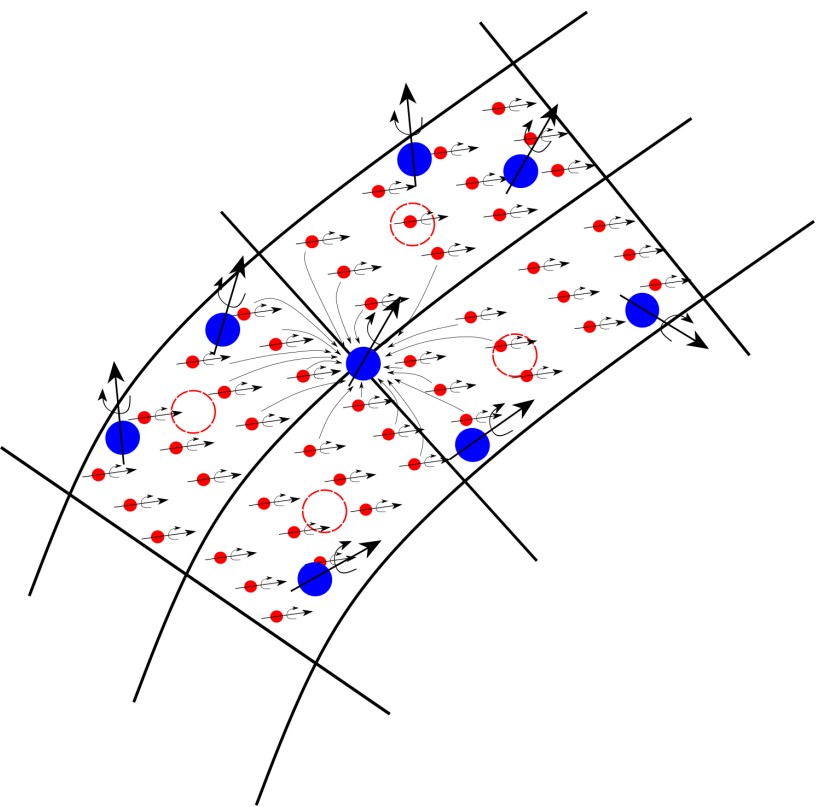

**Figure 3.** The Eulerian solution is used in order to correct the Lagrangian particles that are inside the Eulerian domain. Lagrangian particles are depicted as solid blue circles. Dotted red circles denote the centers of the Eulerian cells. Small solid red circles illustrate the Eulerian particles that correct the Lagrangian ones.

### 2.1. The Eulerian Solver

MaPFlow [41] is an in-house typical Eulerian CFD solver which solves the compressible unsteady Reynolds averaged Navier–Stokes (URANS) equations under a cell-centered finite volume spatial discretization scheme. MaPFlow can handle both structured and unstructured grids; it is parallelized under the MPI protocol, and the grid partitioning is performed using the METIS library [42]. The convective fluxes are evaluated by solving the preconditioned local Riemann problem between the neighboring cells of each face, using the Roe's approximate Riemann solver [43] with the Venkatakrishnan limiter [44]. The viscous fluxes are discretized using a central 2nd order scheme. For the reconstruction of variables at the interface, a piecewise linear interpolation scheme is used. The evaluation of the spatial gradients of the primitive variables is done using the Green–Gauss formula, with a centered scheme approximation. Multiple options are available for turbulence modeling, such as the one-equation model of Spalart–Allmaras [45] or the two-equation model $k - w\ SST$ of Menter [46]. Regarding laminar to turbulent flow transition modeling, the $\gamma - Re_\theta$ model of Menter [47] is used. A delayed detached eddy simulation (DDES) approach is also implemented in MaPFlow, following the suggestions of [48]. Unsteady simulations are performed through an implicit second-order backwards difference scheme [49], along with a dual time-stepping technique [50] in order to facilitate convergence. Finally, the implicit operator inversion is accomplished with the use of the Gauss–Seidel iterative method alongside the reverse Cuthill–Mckee reordering scheme.

### 2.2. The Lagrangian Solver

In a Lagrangian formulation (material coordinates), the flow-field is represented by following the evolution of a number of particles along their trajectories. In that sense,

particles act as flow marker points that are assigned with volume $V_p$ and carry mass $M_p$, dilatation $\Theta_p$, vorticity $\vec{\Omega}_p$ and pressure $\Pi_p$, regarded as the volume integrals of the continuous flow quantities density $\rho$, dilatation $\theta$, vorticity $\vec{\omega}$ and pressure $p$ respectively. In material coordinates, the flow equations take the form:

$$
\begin{aligned}
\frac{d\vec{Z}_p}{dt} &= \vec{U}_p \\
\frac{dV_p}{dt} &= V_p\,\theta_p \\
\frac{dM_p}{dt} &= 0 \\
\frac{d\vec{\Omega}_p}{dt} &= V_p\left[(\vec{\omega}\cdot\nabla)\vec{U} + \frac{1}{\rho^2}\nabla\rho\times\nabla p + \nu\nabla^2\vec{\omega}\right]_p \\
\frac{d\Theta_p}{dt} &= V_p\left[2\|\nabla\vec{U}\| - \frac{1}{\rho}\nabla^2 p + \frac{1}{\rho^2}\nabla\rho\cdot\nabla p + \nu\frac{4}{3}\nabla^2\theta\right]_p \\
\frac{d\Pi_p}{dt} &= V_p\left[(1-\gamma)p\theta + (\gamma-1)\left(\nabla\cdot(\overleftrightarrow{\tau}\cdot\vec{U}) - \vec{U}\cdot(\nabla\cdot\overleftrightarrow{\tau})\right)\right]_p
\end{aligned}
\tag{1}
$$

where $d/dt$ denotes the material time derivative, and $(\cdot)_p$ indicates evaluation at the position of particle $p$. $\nabla\cdot\overleftrightarrow{\tau} = \mu\left(\frac{4}{3}\nabla\theta - \nabla\times\vec{\omega}\right)$ denotes the divergence of the viscous stress tensor, and $\nu = \mu/\rho$ is the kinematic viscosity, which here is assumed constant.

The flow equations are supplemented with the Helmholtz's decomposition theorem (2), which states that every velocity field $\vec{u}$ can be expressed as the sum of a rot-free potential part $\vec{u}_\phi$ and a div-free vortical one $\vec{u}_\omega$, alongside a constant velocity component representing the undisturbed velocity field at infinity $\vec{U}_\infty$. The potential part is defined through a scalar potential $\phi$ $(\vec{u}_\phi = \nabla\phi)$ and is associated with the compressibility effects expressed by the dilatation of the flow $\theta$ $(\theta = \nabla\cdot\vec{u})$, whereas the vortical part is defined through a vector potential (stream-function) $\vec{\psi}$ $(\vec{u} = \nabla\times\vec{\psi})$ which is associated with the free vorticity of the flow $\vec{\omega}$ $(\vec{\omega} = \nabla\times\vec{u})$. Consequently, the scalar and vector potential satisfy the Poisson Equation (3).

$$
\vec{u}(\vec{x},t) = \vec{U}_\infty + \vec{u}_\phi(\vec{x},t) + \vec{u}_\omega(\vec{x},t)
\tag{2}
$$

$$
\begin{aligned}
\nabla^2\phi &= \nabla\cdot\vec{u} = \theta \\
\nabla^2\vec{\psi} &= -\nabla\times\vec{u} = -\vec{\omega}
\end{aligned}
\tag{3}
$$

By using Green's theorem, the velocity field $\vec{u}$ can be expressed in integral form:

$$
\vec{u}(\vec{x}) = \vec{U}_\infty + \int_D (\theta(\vec{y}) + \vec{\omega}(\vec{y})\times)\,\nabla G(\vec{r})\,dD(\vec{y}) + \int_S (\vec{n}\cdot\vec{u}(\vec{y}) + \vec{n}\times\vec{u}(\vec{y})\times)\,\nabla G(\vec{r})\,dS(\vec{y})
\tag{4}
$$

where $G(\vec{r})$ is the Green's function for the Laplace operator, $\vec{r} = \vec{x} - \vec{y}$ and $S = \partial D$. Computational cost is dominated by the convolution integral in (4). For $N$ particles, the associated cost is proportional to $N^2$, which can easily explode as $N$ becomes large and the intended duration of the simulation is long. To make things even worse, when boundaries are present, the surface convolutions in (4) must be also evaluated. In order to reduce computational cost, the PM technique is employed, and the Poisson Equation (3) is solved for the scalar potential $\phi$ and the stream function $\vec{\psi}$. In such a manner, computational cost is minimized from $N^2$ to $N\log N$. Computational performance is also enhanced by using Cartesian grids in order to discretize the Lagrangian domain, thus enabling the employment of fast Poisson solvers [51]. Particularly, in HoPFlow, the James–Lackner algorithm is used [52].

The PM framework is also employed in order to evaluate the right-hand side (RHS) of (1). The Lagrangian particles solution is interpolated to the PM nodes, and the desired differentiations are easily computed through finite difference schemes. Consequently,

the RHS terms are first evaluated on the PM nodes, and then they are interpolated back to the particles positions. Afterwards, time marching is performed through a standard 4th order Runge–Kutta explicit scheme. In every sub-step of Runge–Kutta, intermediate convection steps are carried out, requiring intermediate evaluations of velocity, which are also conducted with the usage of the PM technique.

At the end of every time-step, remeshing is applied in order to recover full coverage of the computational domain and ensure a regular distribution of the numerical particles. Remeshing is a standard procedure in order to prevent excessive concentration or spreading of particles and, in this way, preserve the consistency and accuracy of the numerical solution.

For a given number of particles $\{\vec{Z}_p^n, M_p^n, V_p^n, \vec{\Omega}_p^n, \Theta_p^n, \Pi_p^n\}$, the sub-steps taken in the $n_{th}$ Lagrangian time-step can be listed as follows:

**Step 1:** Project $\{M_p^n, \Theta_p^n, \vec{\Omega}_p^n, \Pi_p^n\}$ on the PM grid and obtain $\rho_{ijk}^n, \theta_{ijk}^n, \vec{\omega}_{ijk}^n, p_{ijk}^n$;

**Step 2:** Solve $\nabla^2 \phi = \theta, \nabla^2 \vec{\psi} = -\vec{\omega}$ and obtain $\phi_{ijk}^n, \vec{\psi}_{ijk}^n, \vec{u}_{ijk}^n$;

**Step 3:** Calculate on the PM grid the RHS terms of (1), e.g., $\nabla \rho_{ijk}^n, \nabla p_{ijk}^n, \nabla \vec{u}_{ijk}^n$;

**Step 4:** Interpolate all grid-based data $q_{ijk}^n$ at the particle positions $q_p^n$;

**Step 5:** Update all particle properties (integrate (1) in time);

**Step 6:** Re-mesh if needed.

### 2.3. The Hybrid Solver

The hybrid solver, HoPFlow, couples the two distinct, previously described, Eulerian and Lagrangian solvers. The Eulerian solution is used only in the regions close to the solid boundaries, whereas the Lagrangian one needs to be valid on the whole computational space, overlapping with the Eulerian one (see Figure 1), in order to satisfy the true far-field boundary conditions.

As mentioned before, the Eulerian and Lagrangian solutions are coupled in two ways. First of all, the Lagrangian part shall undertake to provide the proper flow conditions on the outer boundaries of the Eulerian domain $S_E$. In order to do so, the Lagrangian solution is interpolated from the PM nodes (or, in a more generic approach, from the Lagrangian particle positions) to the ghost cells of the Eulerian grid (see Figure 2). The fluxes at the Eulerian boundary $S_E$ may now be evaluated from the Riemann invariants by taking into account the correct flow information on both sides of $S_E$. Using the correct boundary conditions, MaPFlow shall be capable of describing the flow-field close to the wall boundaries in the detail that is provided by the Eulerian framework.

The closure of the coupling is achieved by correcting the flow information on the PM nodes (more generally on the Lagrangian particles) that lie within the Eulerian domain $D_E$ and then by updating the whole Lagrangian field. In order to do so, the Eulerian solution is transformed into particles that carry mass, dilatation, vorticity, pressure and volume $(\rho, \theta, \vec{\omega}, p, V)_E$. The Eulerian particles need to be densely populated and regularly placed within the Eulerian computational cells, so that full coverage of the PM nodes is ensured. The flow quantities of the Eulerian particles are interpolated from the cell-centered values based on a purely geometric approach using iso-parametric finite element approximations. The presence of solid boundaries is taken into account as surface (singular) particles that carry dilatation $\theta_s$ and vorticity $\vec{\omega}_s$, but no pressure, volume and mass. These particles only affect the solution of the Poisson Equation (3) as contribution in $\vec{n} \cdot \vec{u}(\vec{y})$ and $\vec{n} \times \vec{u}(\vec{y})$ in the surface convolution related to boundary terms (4) and must not be convected during time marching.

The steps followed for the coupling between the Lagrangian and the Eulerian solver are depicted in the flow chart displayed in Figure 4. The corresponding implementation details have been thoroughly analyzed in [3].

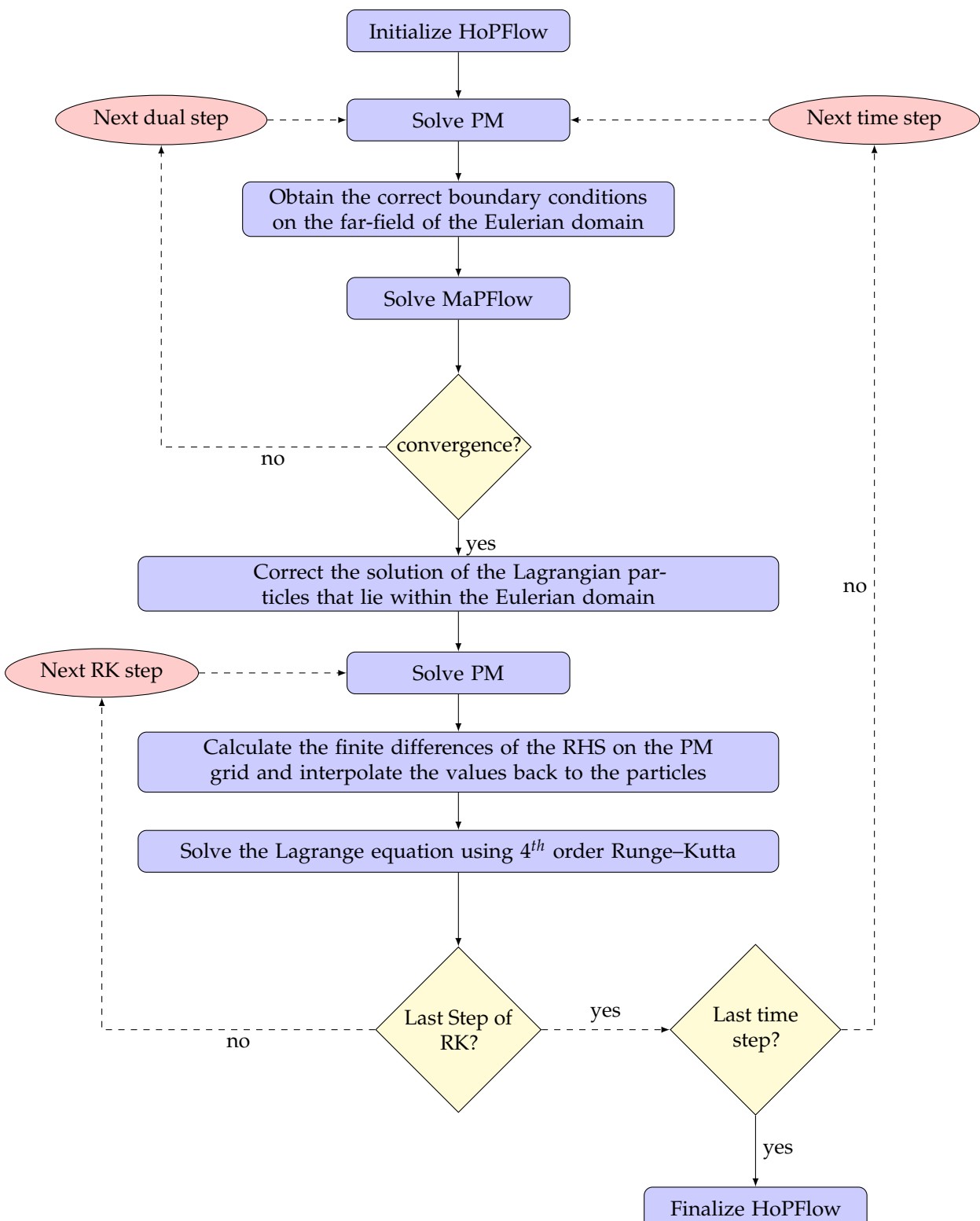

**Figure 4.** Flowchart of the hybrid solver.

## 3. Results

In this section, a detailed overview of the employed numerical parameters is provided. The blades' surface mesh, the time–step value and the PM discretization length are the most significant numerical parameters that can affect the predicted loads. To minimize uncertainties of the produced results, a thorough investigation of these numerical parameters is needed.

### 3.1. Eulerian Solver Results

First, the blades' surface mesh and the time-step value used for the unsteady simulations will be investigated in the Eulerian solver framework. The domain is a cylinder of 20 rotor diameters ($20D$) length, ($5D$ upstream and $15D$ downstream) and $10D$ radius (see Figure 5). In order to take into account the near-wake effect on the aerodynamic loads, the region close to the rotor blades was kept fine. This (blue rectangle in Figure 5) is a cylindrical region that extends up to $1D$ upstream, $3D$ downstream, and $1D$ radially from the rotor hub center, so that the wake expansion is properly accounted for. Furthermore, it needs to be stressed that all these simulations were performed by considering that the whole grid is rotating about the rotor hub center.

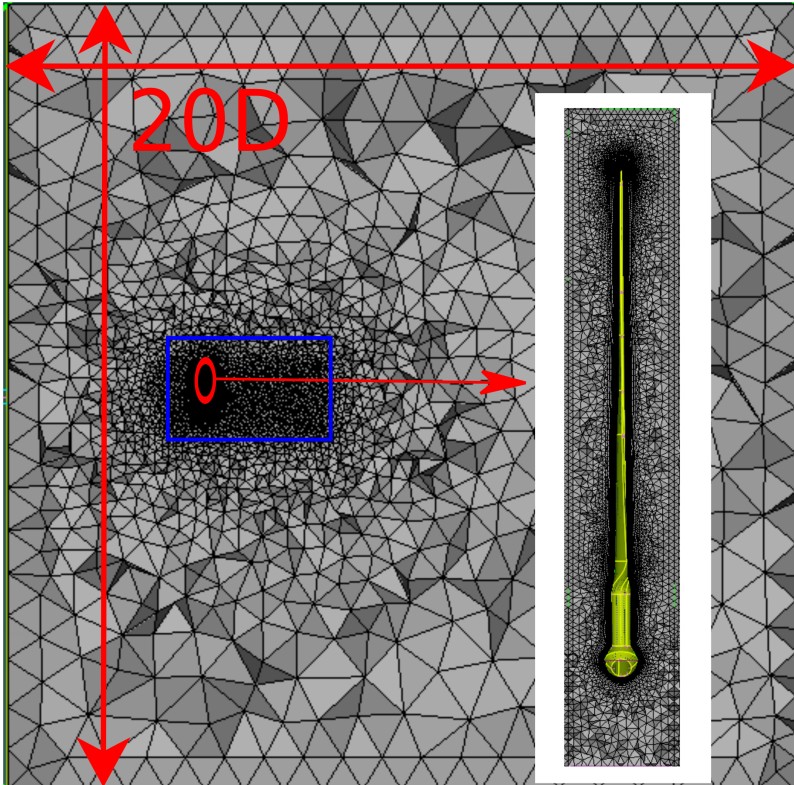

**Figure 5.** Lateral view of the purely Eulerian grid.

### 3.1.1. Blade Surface Mesh Dependency Analysis

Three different blade meshes were tested, employing 5280, 20,350 and 56,260 surface cells for the blade discretization. In the coarse mesh, 66 cells were used to describe the airfoil shape and 80 cells were used in the spanwise direction, whilst in the medium and fine meshes, the corresponding number of cells were $110 \times 185$ and $194 \times 290$, respectively. The corresponding total amount of grid cells are 1.4, 4.8 and 11 million cells, respectively. The integrated rotor loads produced by the three meshes are presented in Table 3. The coarse blade discretization underestimates the thrust value by $\approx 3.8\%$ and the torque by $\approx 13.9\%$ with respect to the finest mesh, whereas the corresponding differences for the medium blade discretization are 2.3% and 3.9%, respectively. Based on the above, it is clear that the medium blade discretization ($110 \times 185$ blade surface cells and 4.8 million total amount of cells) provides the best trade-off between accuracy and computational cost. For this reason, this is the set-up that has been used in the following simulations.

**Table 3.** Thrust and torque estimation with respect to different blade surface meshes. Reference values correspond to finest grid results.

| | Thrust | Torque |
|---|---|---|
| $66 \times 80$ (1.4 *million cells*) | −3.8% | −13.9% |
| $110 \times 185$ (4.8 *million cells*) | −2.3% | −3.9% |
| $194 \times 290$ (11 *million cells*) | 1875.2 [N] | 317.55 [Nm] |

3.1.2. Time-Step Dependency Analysis

In order to investigate how the selected time-step affects the aerodynamic loads, a number of different time-step values ($dt$) have been tested. As one may see in Table 4, the time-step values have been defined with reference to the rotor rotation period ($T$) (certain number of time steps per revolution). It is obvious that all the listed time-step values provide acceptable results as the greatest differences listed are a ≈2% under-estimation of the thrust value at $dt = T/360$ and a ≈1.8% over-estimation of the torque at $dt = T/720$. The final choice of the time-step value for the hybrid solver simulations is also dependent on the grid discretization because of the CFL condition, as pointed out in the following Section 3.2.1.

**Table 4.** Thrust (N) and Torque (Nm) estimation with respect to different time-step values. Reference values correspond to the finest time-step value results.

| | Thrust | Torque |
|---|---|---|
| $dt = T/360$ | −2.0% | +0.3% |
| $dt = T/720$ | −0.1% | +1.8% |
| $dt = T/1440$ | −0.9% | +1.1% |
| $dt = T/2880$ | −0.2% | +1.4% |
| $dt = T/5760$ | 1846.6 [N] | 301.79 [Nm] |

*3.2. Hybrid Solver Results*

In the hybrid solver simulations, the Eulerian sub-domain is restricted to a narrow region around the rotor blades. In particular, it consists of cylinders that surround the rotor blades and extend at least up to 1 local chord away from the largest section of the blade, as is recommended in [3] and illustrated in Figure 6. The greatest chord length of the specific blade is approximately 240 mm at 20% of its radius. The Eulerian mesh consists of hexahedral cells (structured-type) close to the blade surface in order to better represent the boundary layer properties, whilst it is unstructured at the rest of the domain. Another numerical parameter that needs to be considered is that the largest dimension of the Eulerian cells should not exceed the PM discretization length. Otherwise, the density of Eulerian particles (particles generated within the Eulerian grid) may not be sufficient to ensure full coverage of the PM nodes. Typically, this restriction concerns the surface discretization of the Eulerian sub-domain far-field ($S_E$); however, care needs to be taken of the span-wise discretization of the blade as well. This justifies the great number of cells used in the span-wise direction of the blade surface meshes that were tested in Section 3.1.1. The Eulerian sub-domain that is used in the hybrid solver simulations consists of 4.1 millon computational cells.

The Lagrangian sub-domain is defined as a box that covers the entire computational domain, extending from $1D$ upstream up to $2.5D$ downstream and $1D$ radially about the rotor hub center. In Figure 7, the placement and the extent of the two sub–domains is depicted. As stated in Section 2.2, the Lagrangian sub–domain is discretised with the use of the PM technique and by employing uniform Cartesian grids. The different values of the PM discretisation length ($DXpm$) that are tested have been chosen to vary proportionally to the local chord length at 75% of the blade radius, which is approximately 120 mm and from now on will be denoted by $c$.

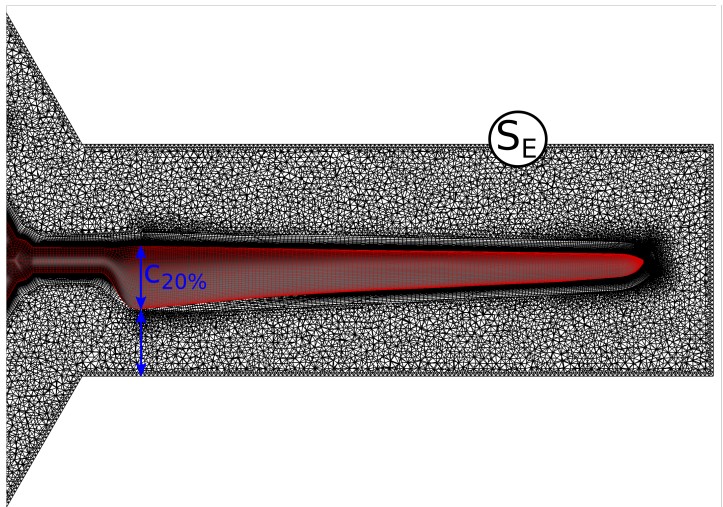

**Figure 6.** Eulerian sub-domain used in hybrid solver simulations.

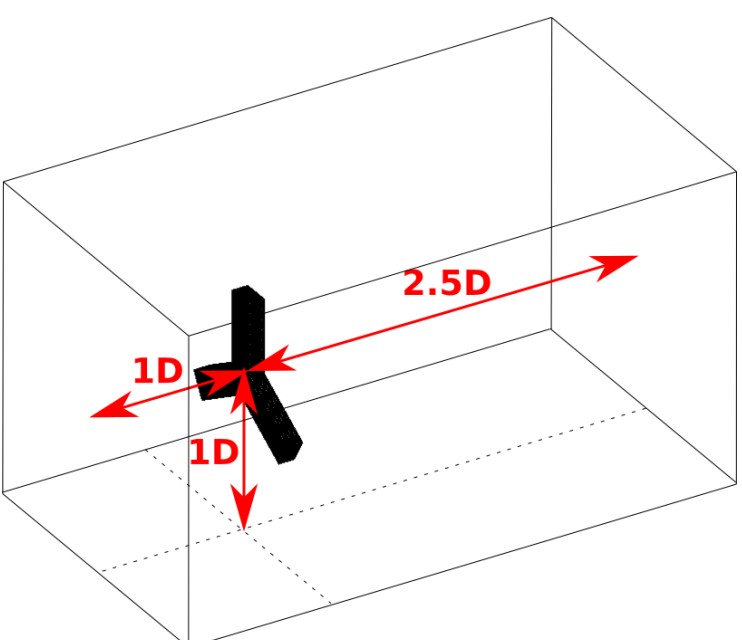

**Figure 7.** Visualization of the Lagrangian and the Eulerian sub-domains in the hybrid solver simulations.

### 3.2.1. PM Grid Dependency

In Table 5, the integrated rotor loads predicted for different values of $DXpm$ are listed. Five different values of $DXpm$ have been tested, $DXpm = 1c$, $DXpm = 0.5c$, $DXpm = 0.35c$, $DXpm = 0.25c$ and $DXpm = 0.20c$, which correspond to 2, 3 , ≈4, 5 and 6 PM nodes per chord, respectively. It needs to be stressed that the time-step values employed in these simulations correspond to more than 360 steps per rotor revolution, complying with the results in Section 3.1.2. Nevertheless, for the hybrid solver simulations, there is one extra restriction. Since the time-marching scheme is explicit, the time-step values need to respect the CFL condition. Consequently, $dt = T/540$, $dt = T/900$, $dt = T/1440$, $dt = T/1800$ and $dt = T/2160$ have been utilized for the $DXpm = 1c$, $DXpm = 0.5c$, $DXpm = 0.35c$, $DXpm = 0.25c$ and $DXpm = 0.20c$ simulations, respectively.

It is clearly shown in Figure 8 that the differences in predicted rotor loads decrease as $DXpm$ gets smaller, with values less than $DXpm = 0.35c$ (at least 4 points per chord length) providing a PM grid independent solution. Apart from the integrated rotor loads, the detailed description of the radial distribution of the aerodynamic loads is also of great

importance. Table 6 shows the normalized forces and moments at 60% of the blade, predicted by the different values of $DXpm$. The loads of the specific radial position experience the highest sensitivity with respect to numerical parameters (which will be also shown in Section 3.2.2). For this reason, the rest of the available span-wise positions are neglected in this table. It is obvious that in order to obtain the blade radial distribution of aerodynamic loads in detail, at least 5 PM nodes per chord length ($DXpm \leq 0.25c$) need to be used. However, it also needs to be highlighted (see Table 5) that as $DXpm$ gets smaller, the total number of PM nodes increases dramatically, thus substantially penalizing computational cost. Based on all the above remarks, $DXpm = 0.25c$ seems to provide the best compromise between accuracy and computational cost.

**Table 5.** Thrust (N) and Torque (Nm) estimation of different PM discretisation lengths. Reference values correspond to minimum value $DXpm = 0.20c$.

| DXpm | dt | PM Nodes | PM Nodes per $c$ | Thrust | Torque |
|------|----|----------|------------------|--------|--------|
| $1c$ | $T/540$ | 1.1 million | 2 | +17.7% | +53.8% |
| $0.50c$ | $T/900$ | 7.3 million | 3 | +4.1% | −1.3% |
| $0.35c$ | $T/1440$ | 23.3 million | 4 | +0.5% | +0.7% |
| $0.25c$ | $T/1800$ | 52.2 million | 5 | +0.03% | +0.25% |
| $0.20c$ | $T/2160$ | 123.2 million | 6 | 1796.3 [N] | 293.6 [Nm] |

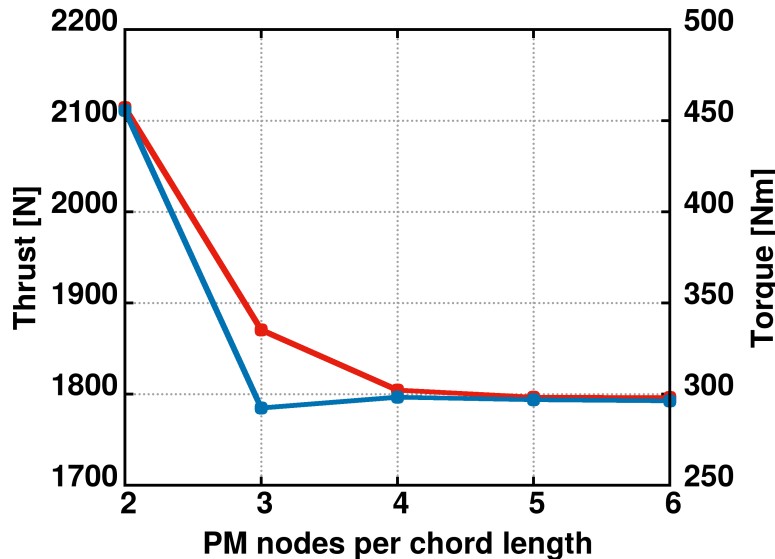

**Figure 8.** Thrust (N) and Torque (Nm) estimation with respect to the number of PM nodes per chord length.

**Table 6.** Normalized aerodynamic load estimation at 60%$R$ provided by different PM discretisation lengths. Reference values correspond to minimum value $DXpm = 0.20c$.

| $DXpm$ | $F_N/dr$ (60%$R$) | $F_T/dr$ (60%$R$) | $M_{tw}/dr$ (60%$R$) |
|--------|-------------------|-------------------|----------------------|
| $1c$ | +16.06% | +50.42% | +18.44% |
| $0.50c$ | +4.15% | −4.57% | +8.29% |
| $0.35c$ | +0.98% | +5.81% | +1.03% |
| $0.25c$ | +0.42% | +1.24% | −1.27% |
| $0.20c$ | 356.8 [N/m] | 35.44 [N/m] | 4.876 [Nm/m] |

### 3.2.2. Comparison against Measurements

In this Section, the simulation results by the Eulerian solver, MaPFlow, and the hybrid solver, HoPFlow (by using $DXpm = 0.25c$), are compared against available experimental data. It needs to be stressed that the total amount of computational elements (PM nodes

and cells of the Eulerian sub–domain) used in the hybrid solver simulations are about an order of magnitude more than the computational cells used in the Eulerian simulation. In Table 7, the measured and computationally predicted rotor loads are listed. The hybrid solver seems to predict a thrust value that is closer to the measured value (overestimated by ≈9%), as compared with the Eulerian solver (overestimated by ≈13%). Better agreement with measurements is achieved in torque prediction by both computational tools (HoPFlow under-predicts the torque value by ≈7% and MaPFlow by ≈4%). In Table 8 and Figure 9, the normal and tangential forces distribution along the blade span are depicted. Overall, simulations predict higher values of the aerodynamic forces on most radial positions, with the hybrid solver results being closer to measurements than its Eulerian counterpart. This is attributed to the increased numerical diffusion of the Eulerian methodology resulting from the gradual coarsening of the computational grid towards the far-field. Consequently, the wake is dissipated when convected downstream, and its upstream induced effect (downwash) is not properly resolved, yielding in over-estimation of aerodynamic loads. On the other hand, the Lagrangian formulation of the Navier–Stokes equations and the usage of vorticity as the primary flow quantity of the particles reduces numerical diffusion. Consequently, the near-wake deficit is effectively preserved, yielding a more physical representation of the flow-field [32]. This results in the prediction of increased axial induction and thus reduced loads. The maximum discrepancies with respect to the experimental values are observed about the mid-span of the blade (60%), where a ≈22% and ≈18% higher normal force is predicted by the Eulerian and the hybrid solver, respectively. The corresponding differences in the tangential force are ≈28% and ≈21%. Even though the percentage differences concerning normal forces at the root of the blade (25% and 35%) are slightly bigger, they are not considered that important. The high percentage difference comes from the small reference value, whereas the absolute error in the loads estimation is quite smaller. Much better agreement is achieved at the tip region (82% and 92%), where again the hybrid solver results are closer to the measured values as compared with those of its Eulerian counterpart.

In Figure 10, the gauge pressure distribution of the Eulerian and hybrid solvers at various radial positions is compared against experimental measurements. Discrepancies with respect to measured data are observed in both computational tools results at the root region, which, however, seem to agree well with each other. A minor level shift towards higher pressure is predicted by the hybrid solver on the pressure side of the blade. Nevertheless, the predicted pressure difference between the pressure and suction side seem to be comparable (area under the pressure plots of the two sides), which explains the small differences in the corresponding loads, shown in Table 8 and Figure 9. On the contrary, the pressure side predictions of the computations agree very well with each other and with the experimental values at 60% of the blade. However, the suction side pressure close to the leading edge is over-predicted by both MaPFlow and HoPFlow, with the latter being slightly closer to the measured data. This explains why the maximum differences between predicted and measured forces are observed in the mid-span of the blade, as stated before. On the other hand, very good agreement is observed at the tip region. Both the pressure and suction side distributions are very close to each other, which is in line with the very well-predicted normal forces at 82% and 92% of the blade.

To sum up, predictions and measurements agree very well with each other closer to the tip region. The tip region loading dictates the overall loading and performance of the rotor. This explains why the differences in the integrated rotor loads are not substantial, as shown in Table 7. The slightly smaller values of aerodynamic forces (closer to the experimental ones) predicted by the hybrid solver are attributed to the reduced numerical diffusion and thus the increased wake induction. Small discrepancies are shown in the root region that seem to be more pronounced in the middle part of the blade. This may be explained by the fact that the spinner geometry is not included in the computations. As a result, root vortices are emitted in the simulations that tend to have a significant effect on the local flow of the root sections [32].

**Table 7.** Thrust (N) and Torque (Nm) estimation. Comparison between computational predictions and experimental measurements. Reference values correspond to measured values.

|  | Thrust | Torque |
| --- | --- | --- |
| MaPFlow | +13% | −4% |
| HoPFlow | +9% | −7% |
| measurements | 1620.1 [N] | 319.33 [Nm] |

**Table 8.** Radial distribution of normal and tangential forces. Comparison between experimental measurements and computational predictions. Reference values correspond to experimental measured values.

|  | $F_N$ [N/m] | | | $F_T$ [N/m] | | |
| --- | --- | --- | --- | --- | --- | --- |
| Radius | Measurements | MaPFlow | HoPFlow | Measurements | MaPFlow | HoPFlow |
| 25% | 119.0 | +29% | +25% | 31.98 | −11% | −14% |
| 35% | 198.1 | +8% | +7% | 27.23 | +21% | +14% |
| 60% | 303.4 | +22% | +18% | 29.76 | +28% | +21% |
| 82% | 421.9 | +6% | +3% | 44.22 | +7% | +2% |
| 92% | 452.9 | +1% | −1% | 43.37 | +7% | +4% |

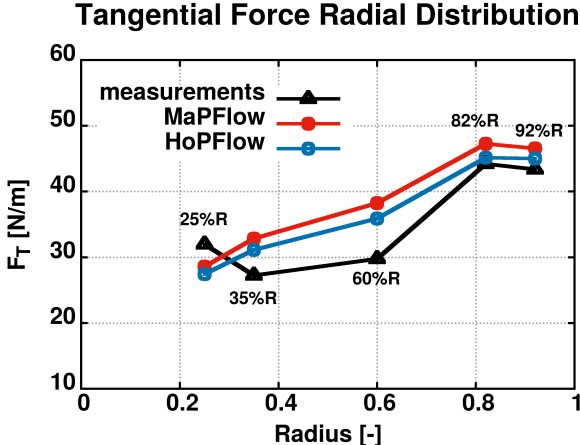

**Figure 9.** Radial distribution of normal and tangential forces. Comparison between experimental measurements and computational predictions.

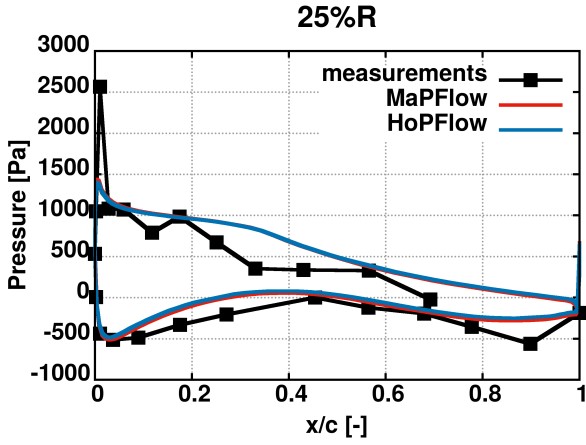

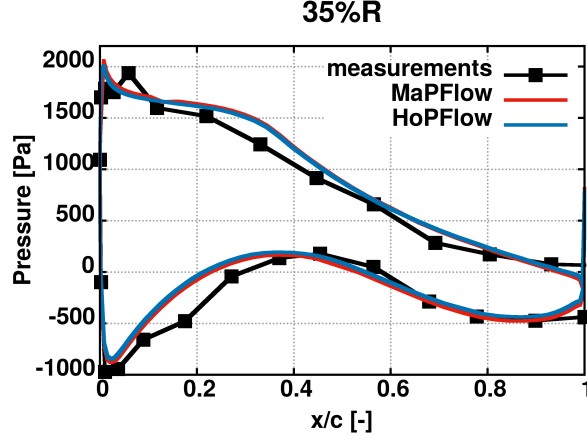

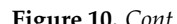

**Figure 10.** *Cont.*

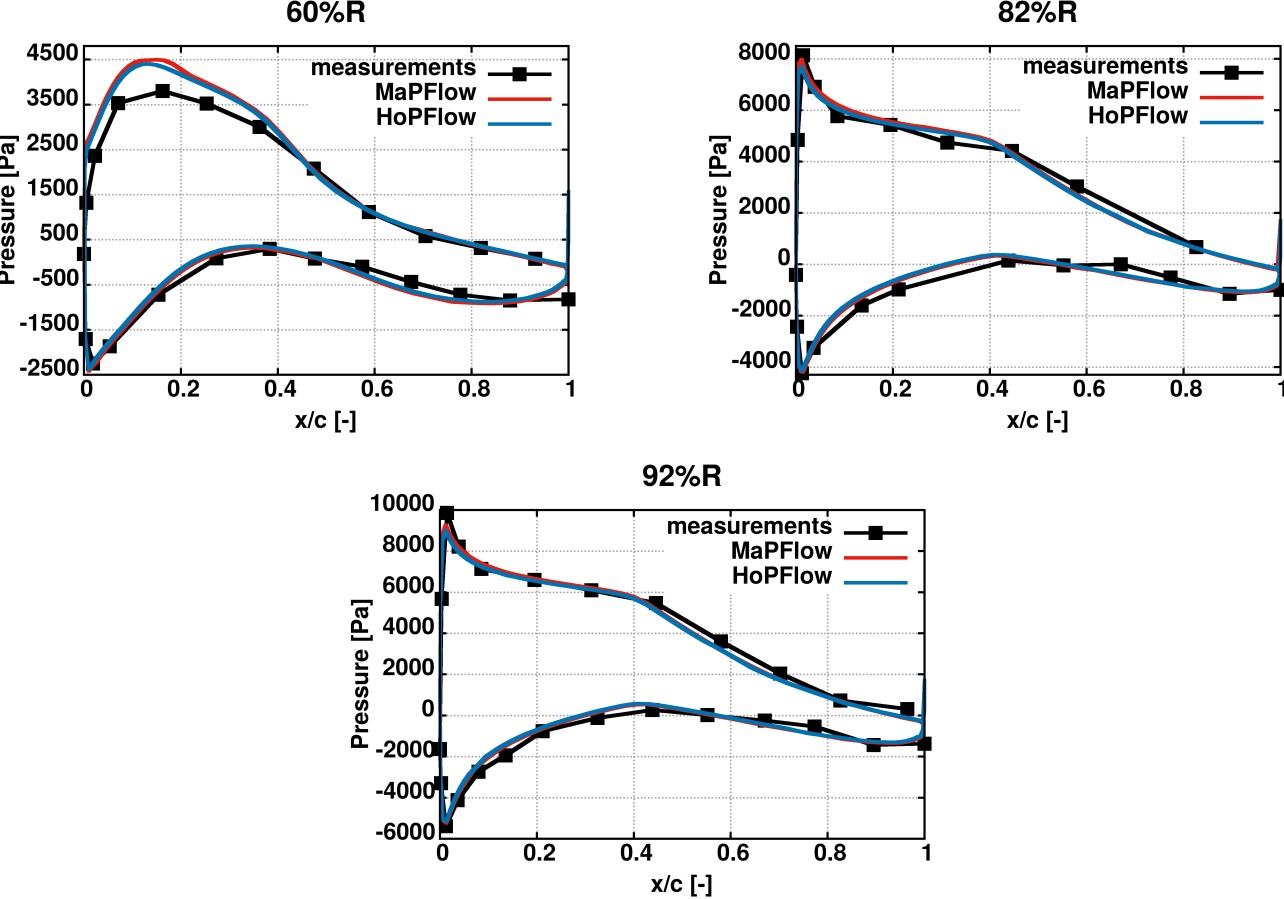

**Figure 10.** Pressure distribution. Comparison between experimental measurements and predictions by different computational tools.

### 3.2.3. Computational Requirements

In Table 9, a comparison concerning the computational cost of the Eulerian and two hybrid solver simulations is made. In the former, a coarse PM grid ($DXpm = 0.50c$) has been used, whereas in the latter, the fine PM grid ($DXpm = 0.25c$) has been used, in which the grid dependency analaysis (Section 3.2.1) resulted. It is noted that the PM grids for which computational costs are presented result in similar forces predictions, within a 5% margin (see Table 6). Due to the usage of uniform PM grids, even the coarse discretization length ends up in approximately 2 times more computational elements than the ones used in the Eulerian solver simulation, which rises to 12 times if the fine discretization length is employed. Nevertheless, the Eulerian solver simulation needs more rotor revolutions in order to achieve convergence of aerodynamic loads. This is because the Lagrangian method exactly satisfies the boundary conditions at infinity, and therefore, the required number of revolutions for the convergence of the loads depends only on the distance that the wake has travelled away from the rotor disk plane. On the other hand, in the Eulerian simulation, the convergence rate depends on the extent of the domain, which dictates the reflection of the numerical errors and the coarsening of the grid in the far-field, which regulates their decay rate. The differences in the utilized time-step values (expressed as steps per rotor revolution) come from the fact the CFL condition needs to be respected in the hybrid simulations, as no preconditioning has been applied to the Lagrangian formulation of the flow Equation (1). This explains the greater number of steps per revolution required by the fine grid simulation. Even though the amount of PM nodes used in the fine grid simulation is an order of magnitude greater than the ones used in the coarse grid simulation, the computational time-lengths of the time-steps are comparable. This is due to the fact that

in the coarse grid simulation, more sub-iterations are needed to accomplish a converged time-step solution. Furthermore, fewer processors have been used in the specific simulation. Overall, the computational cost of the coarse PM grid hybrid simulation is 1.4 times higher than that of the Eulerian solver simulation, whereas the fine PM grid hybrid simulation costs 5.7 times more than the Eulerian one.

In Figure 11, the normal force distribution computed by the three different simulations is depicted. The distributions predicted by the Eulerian and the coarse grid hybrid simulation are close to each other, whereas the distribution predicted by the fine grid hybrid simulation is slightly closer to the experimental values. In Figure 12, the pressure distribution at 60% of the blade is depicted. The coarse grid hybrid simulation results exhibit a small shift towards lower pressure with respect to measured data–sets and other predictions, which results from the insufficient density of the PM grid used. However, these deviations in pressure do not significantly affect the overall aerodynamic force prediction.

**Table 9.** Computational cost comparison between Eulerian and hybrid solver simulations.

| Solver | Computational Elements | Revolutions | Steps Revolution | Sec Step | Sub-Iterations Step | Processors | Corehours |
|---|---|---|---|---|---|---|---|
| **MaPFlow** | $4.8M$ [a] | 10 | 1440 | 3.5 | 9 | 480 | 6720 |
| **HoPFlow$_{coarse}$** | $7.3M$ [b] $+ 4.1M$ [a] | 6 | 900 | 76.75 | 17 | 80 | 9210 |
| **HoPFlow$_{fine}$** | $52.2M$ [b] $+ 4.1M$ [a] | 6 | 1800 | 81 | 9 | 120 | 38,800 |

[a] Number of computational cells. [b] Number of PM nodes.

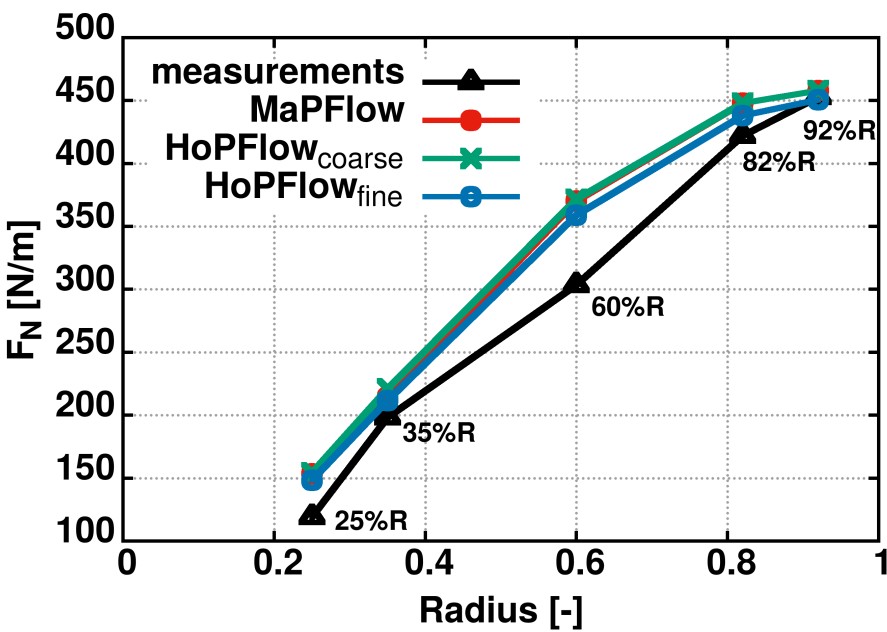

**Figure 11.** Radial distribution of normal forces. Comparison between experimental measurements and computational predictions.

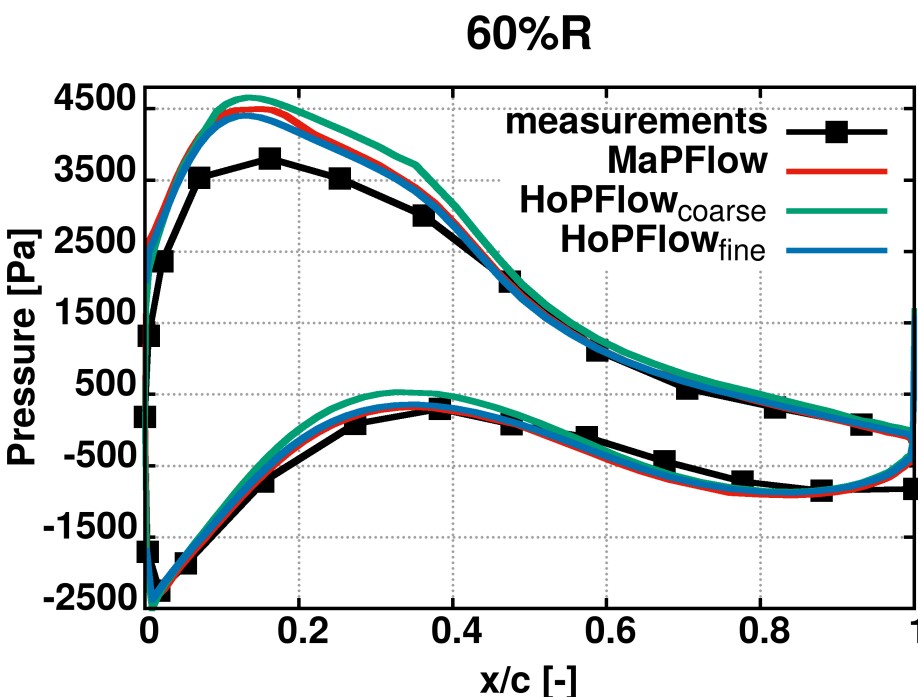

**Figure 12.** Pressure distribution at 60% of the blade. Comparison between experimental measurements and computational predictions.

## 4. Conclusions

In this paper, the hybrid Lagrangian–Eulerian solver HoPFlow has been presented, along with details about its distinct solvers and the coupling between them. The Eulerian solver, MaPFlow, solves the compressible Navier–Stokes equations under a cell-centered finite-volume discretization scheme, while the Lagrangian solver uses numerical particles that carry mass, pressure, dilatation and vorticity as flow markers in order to represent the flow-field by following their trajectories. The calculation of the velocity field is performed by utilizing the decomposition theorem introduced by Helmholtz. Computational performance is enhanced by utilizing the particle mesh (PM) methodology in order to solve the Poisson equations for the scalar potential $\phi$ and the stream function $\vec{\psi}$.

The hybrid solver is tested in 3-D unsteady simulations concerning the axial flow around the wind turbine model rotor used in the New MEXICO experimental campaign. Simulation results are presented as integrated rotor loads, span-wise distribution of aerodynamic loads and gauge pressure distribution at various span-wise positions along the rotor blades. Comparison is made against experimental measured data and computational results produced with the Eulerian solver.

Under a PM discretisation with 5 nodes per chord length (in this analysis the characteristic chord length is assumed to be the on at 75% of the blade), the hybrid solver predicts the loading close to the blade tip with great accuracy. This is regarded to be very important, as the aerodynamic loads close to the blade tip tend to dominate the whole rotor performance and the dynamic excitation of the blades in aeroelastic simulations. Deviations from the experimentally measured values are observed close to the blade root, which become more pronounced in the middle region of the blade. However, very good agreement is shown with the corresponding Eulerian solver results. These discrepancies are attributed to the absence of the spinner geometry in both hybrid and Eulerian solver simulations. Furthermore, it needs to be highlighted that the Lagrangian formulation used for the wake description, alongside with employment of vorticity as the primary flow quantity of the particles, reduces numerical diffusion significantly compared to the Eulerian solver. For this reason, the wake is resolved much more efficiently in the hybrid solver simulations, thus resulting in greater wake-induced velocities. Consequently the hybrid

solver ends up with a better estimation of the aerodynamic loads along the whole blade span that is closer to the experimental values.

Summarizing the above, the results presented in this work indicate that the accuracy of the boundary layer solution (near-body flow-field) of the hybrid solver is comparable with that produced by a standard Eulerian CFD code. This confirms that the coupling method that determines the boundary conditions for the confined Eulerian grid is adequate and consistent. Nevertheless, the cost of the hybrid approach is overwelming for a single rotor simulation. A remedy for moderating the computational cost is the application of multi-level/telescopic PM grids [15], finer in the vicinity of the body and coarser in the far-domain. The benefit for paying this increased computational cost (using fine PM grids in the entire computational domain) lies in unsteady applications where interaction phenomena (e.g. rotor-rotor interactions) are prominent or in cases where the accurate characterization of the far-wake dynamics is required. It is also attractive for aeroelastic analyses in which a standard Eulerian methodology relies on the use of overset grids and the corresponding coupling between different sub-domains, which penalizes computational cost.

**Author Contributions:** Conceptualization, G.P. and V.A.R.; Methodology, G.P.; Resources, V.A.R.; Software, N.S., G.P. and V.A.R.; Supervision, G.P., J.M.P. and V.A.R.; Validation, N.S.; Visualization, N.S.; Writing—original draft, N.S.; Writing—review & editing, N.S., G.P., J.M.P. and V.A.R. All authors have read and agreed to the published version of the manuscript.

**Funding:** The research work was supported by the Hellenic Foundation for Research and Innovation (HFRI) under the HFRI PhD Fellowship grant (Fellowship Number: 797. This research is co-financed by Greece and the European Union (European Social Fund-ESF) through the Operational Program «Human Resources Development, Education and Lifelong Learning» in the context of the project "Strengthening Human Resources Research Potential via Doctorate Research—2nd Cycle" (MIS-5000432), implemented by the State Scholarships Foundation (IKY).

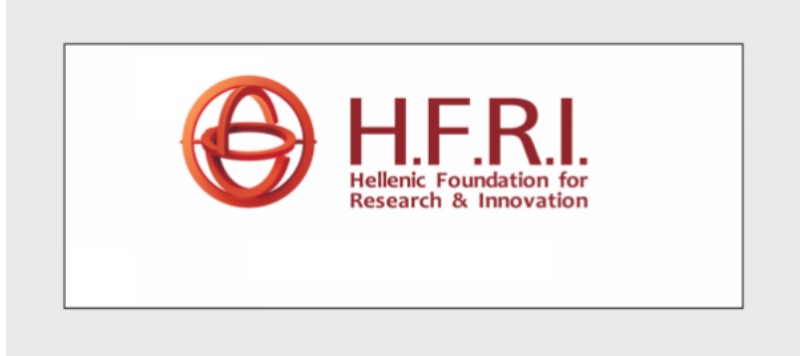

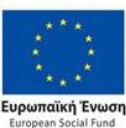 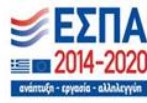

**Data Availability Statement:** Data presented in the paper are free to share upon request.

**Acknowledgments:** NTUA computations were supported by computational time granted from the Greek Research & Technology Network (GRNET) in the National HPC facility ARIS under projects "ELASTODYN" with ID pr010013_thin and "HELIHOP" with IDs pr012024_thin and pr012024_fat. The computational grids were generated using the ANSA CAE pre-processor of "BETA_CAE Systems S.A.".

**Conflicts of Interest:** The authors declare no conflict of interest.

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
