# Peer review of "Assessment of a Hybrid Eulerian–Lagrangian CFD Solver for Wind Turbine Applications and Comparison with the New MEXICO Experiment"

_fluids, doi:10.3390/fluids7090296_

Round 1

Reviewer 1 Report

The manuscript presents the results of the application of a hybrid Eulerian-Lagrangian CFD solver to the aerodynamics simulation of the New MEXICO wind turbine experiment.

The manuscript is well written and clear in both the definition of its background and aims as well as the description of the results obtained. Minor improvements could be made in the description of the coupling between the Eulerian solver and the Lagrangian one as well as the description of the numerical setup used for the numerical experiment: in particular it is not totally clear the "relative" weight between the Eulerian domain and the Lagrangian one, a figure could make this clear.

The results presented are scientifically correct and although they are not really novel, they are still of interest.  However, the results could be more interesting if also the computation costs are presented: the accuracy differences between the Eulerian results and the hybrid one are evident, but these differences seem to be "not resounding". To provide the right relevance to the hybrid solver results, they should be accompanied by their computational costs: in particular, it is not clear if the Eulerian solver could match the same accuracy of the hybrid solver by means of a more refined grid with lower computational costs than the hybrid setup. The computational costs comparison could clarify this aspect.

Author Response

Point 0: The manuscript presents the results of the application of a hybrid Eulerian-Lagrangian CFD solver to the aerodynamics simulation of the New MEXICO wind turbine experiment.

First of all we want to thank the reviewer for his careful reading and his valuable comments that helped us improve our work!

Point 1: The manuscript is well written and clear in both the definition of its background and aims as well as the description of the results obtained. Minor improvements could be made in the description of the coupling between the Eulerian solver and the Lagrangian one as well as the description of the numerical setup used for the numerical experiment: in particular it is not totally clear the "relative" weight between the Eulerian domain and the Lagrangian one, a figure could make this clear.

Response 1: In order to make the coupling between the Eulerian and the Lagrangian solver comprehensible, a flow-chart of the hybrid solver has been added in Section 2.3. In order to clarify the relative weight between the Lagrangian and the Eulerian sub-domains, Figure 7 has been added to Section 3.2, illustrating the two sub-domains placement and extent. Furthermore, in the same Section, an explicit statement of the computational cells of the Eulerian sub-domain used in the hybrid simulations has been added, whereas in Table 5 the number of the Particle Mesh nodes used in these simulations is listed. Finally, the relative difference in the total amount of computational elements between the hybrid solver and the Eulerian solver simulations has been stressed in Section 3.2.2.

Point 2: The results presented are scientifically correct and although they are not really novel, they are still of interest. However, the results could be more interesting if also the computation costs are presented: the accuracy differences between the Eulerian results and the hybrid one are evident, but these differences seem to be "not resounding". To provide the right relevance to the hybrid solver results, they should be accompanied by their computational costs: in particular, it is not clear if the Eulerian solver could match the same accuracy of the hybrid solver by means of a more refined grid with lower computational costs than the hybrid setup. The computational costs comparison could clarify this aspect.

Response 2: In order to provide the reader with the right relevance between the hybrid solver and the Eulerian solver results, Section 3.2.3 has been added in the manuscript, in which the computational cost details of the two solvers are compared. An Eulerian simulation with a more refined grid (LES type simulation with the same amount of computational elements as the ones used in the hybrid solver simulation) could not be perfomred in the given time-frame. However, a fair comparison is now made between the Eulerian simulation and a coarse Particle Mesh grid hybrid simulation, which has a similar computational cost with the Eulerian solver simulation. In this way we believe to assess both the differences in computational cost when results of comparable accuracy are produced and the differences in the accuracy of the produced results when computational costs are comparable. To enforce the statements made in this section, the last paragraph in the Conclusions and the penultimate paragraph of the Introduction have been properly adjusted.

PS: Minor revisions have been made throughout the whole text (mainly in Introduction and Methodology) in order to avoid similarity issues.

Reviewer 2 Report

A strongly coupled in–house hybrid solver, named HoPFlow, is presented and assessed in WT applications- can the solver be make available in open source? Otherwise, there is not much use of this work other than citation.

https://github.com/MaPFlow/vpm_free - is this the MapFlow used? If so, please add this as a reference and identify the version used.

Add a few newer references like https://www.mdpi.com/2311-5521/6/12/460 ("Lagrangian vs. Eulerian: An Analysis of Two Solution Methods for Free-Surface Flows and Fluid Solid Interaction Problems") .

Author Response

First of all we want to thank the reviewer for his careful reading and his valuable comments that helped us improve our work!

Point 1: A strongly coupled in–house hybrid solver, named HoPFlow, is presented and assessed in WT applications- can the solver be make available in open source? Otherwise, there is not much use of this work other than citation.

Response 1: Unfortunately the solvers cannot be made available open source, but they are free to share upon request for academic use. In order to make clear to the reader the value of this work, the motivation of the work is now clearly stated in the penultimate paragraph of the Introduction.

Point 2: https://github.com/MaPFlow/vpm_free - is this the MapFlow used? If so, please add this as a reference and identify the version used.

Response 2: This is an obsolete MaPFlow version that is no longer in use.

Point 3: Add a few newer references like https://www.mdpi.com/2311-5521/6/12/460 ("Lagrangian vs. Eulerian: An Analysis of Two Solution Methods for Free-Surface Flows and Fluid Solid Interaction Problems").

Response 3: The specific reference has been added in the Introduction as an SPH reference.

PS: Minor revisions have been made throughout the whole text (mainly in Introduction and Methodology) in order to avoid similarity issues.

Reviewer 3 Report

The manuscript has the potential to advance our understanding on combined Eulerian-Lagrangian CFD solver and their accuracy. Many of the author's arguments, however, require additional citations. Several examples are provided below:

1. Please add schematic diagrams to sections 2.1 Eulerian Solver and 2.2 Lagrangian Solver.

2. Please provide a detailed explanation of the numerical scheme used to solve proposed Hybrid Solver along with schematic diagrams. I recommend that the authors provide the following references: (a) Sarker, S. (2022) Essence of MIKE 21C (FDM Numerical Scheme): Application on the River Morphology of Bangladesh. Open Journal of Modelling and Simulation, 10, 88-117. doi: 10.4236/ojmsi.2022.102006, (b) Sarker, S. (2022) A Short Review on Computational Hydraulics in the Context of Water Resources Engineering. Open Journal of Modelling and Simulation, 10, 1-31. doi: 10.4236/ojmsi.2022.101001.

3. Please provide a detailed description of your experimental setup.

Author Response

Point 0: The manuscript has the potential to advance our understanding on combined Eulerian-Lagrangian CFD solver and their accuracy. Many of the author's arguments, however, require additional citations. Several examples are provided below:

First of all we want to thank the reviewer for his careful reading and his valuable comments that helped us improve our work!

Point 1: Please add schematic diagrams to sections 2.1 Eulerian Solver and 2.2 Lagrangian Solver.

Response 1: The Eulerian solver is a typical CFD solver under a finite volume discretization scheme. This is now clearly stated in section 2.1 and alongside with the given references the authors believe that there are no more gray areas to be clarified. A list of all the sub-steps taken in every Lagrangian solver step is now added at the end of Section 2.2 to increase the clarity of the specific section as well.

Point 2: Please provide a detailed explanation of the numerical scheme used to solve proposed Hybrid Solver along with schematic diagrams. I recommend that the authors provide the following references: (a) Sarker, S. (2022) Essence of MIKE 21C (FDM Numerical Scheme): Application on the River Morphology of Bangladesh. Open Journal of Modelling and Simulation, 10, 88-117. doi: 10.4236/ojmsi.2022.102006, (b) Sarker, S. (2022) A Short Review on Computational Hydraulics in the Context of Water Resources Engineering. Open Journal of Modelling and Simulation, 10, 1-31. doi: 10.4236/ojmsi.2022.101001.

Response 2: A flow-chart of the hybrid solver has been added in Section 2.3, in order to make clear the numerical details of the proposed solver.

The recommended by the reviewer references fall within the broad area of computational hydraulics-fluid mechanics. The methods described in the papers are standard Eulerian CFD methodologies, while the focus of the present work lies in the use of Lagrangian formulations and their coupling with Eulerian ones. Therefore, we don’t see a strong relevance between these references and the present work. However, we do recognize the correctness of the reviewer’s point. For this reason a few more references have been added concerning particle methodologies applied on Wind Turbine applications and numerical investigations of the New MEXICO experimental cases performed by other researchers. Finally, 3 classical books concerning CFD have been added as references in Eulerian methodologies.

Point 3: Please provide a detailed description of your experimental setup.

Response 3: More details of the experimental set-up have been provided in the last paragraph of the Introduction, according to the accompanying references.

PS: Minor revisions have been made throughout the whole text (mainly in Introduction and Methodology) in order to avoid similarity issues.

Round 2

Reviewer 3 Report

Thanks for the revision.